# CAN SYMBOLIC REGRESSION OF BOOLEAN FUNCTIONS BOOST LOGIC SYNTHESIS?

## ABSTRACT

Logic synthesis, which aims to synthesize a *compact* logic circuit with minimized size while *exactly* satisfying a given functionality, plays an important role in chip design. Recently, symbolic regression (SR) has shown great success in scientific discovery to recover underlying mathematical functions from given datasets. However, we found from extensive experiments that existing SR methods struggle to recover an *exact* and *compact* boolean function for logic synthesis given a truth table, i.e., complete input-output pairs of the circuit. The major challenges include (1) the greater complexity of underlying boolean functions compared to mathematical functions, and (2) the complex objectives involving both exact recovery and expression optimization towards circuit minimization. To address these challenges, we propose a novel **s**ymbolic factor**i**zed boolea**n** s**e**archer (SINE) to recover exact and compact boolean functions towards logic synthesis. Motivated by the Shannon decomposition theorem, SINE proposes a factorized boolean function representation to decompose the underlying boolean function into multiple simplified sub-functions, significantly reducing their complexity and thus improving the recovery accuracy. Moreover, based on the key observation that, logical sharing is significant for circuit size minimization. SINE proposes a self-symmetric sub-expression motif operators mining mechanism to enhance the monte-carlo tree search method for optimized boolean function learning. To the best of our knowledge, SINE is *the first* symbolic regression framework capable of exactly recovering optimized boolean functions for circuit optimization. Experiments on circuits across a wide range of inputs demonstrate that SINE significantly improves the recovery accuracy and decreases the size of synthesized circuits by up to 24.32% compared to state-of-the-art methods.

## 1 INTRODUCTION

Complex integrated circuits (ICs) can contain billions of transistors, making manual design infeasible (Huang et al., 2021). Consequently, the IC industry depends on electronic design automation (EDA) tools (Wang et al., 2009), which systematically transform high-level hardware descriptions into layouts ready for IC fabrication. A critical step in this process is logic synthesis (LS), which converts a behavioral-level description of a design into an optimized gate-level circuit. The primary goal of LS is to minimize the delay and area of the circuit. As LS is the initial step in EDA processes that produce the final IC layout, the quality of its output significantly impacts the area, power, and performance of the IC (De Abreu et al., 2021; Berndt et al., 2022).

Logic synthesis (LS) is a challenging $\mathcal{NP}$-hard combinatorial optimization problem. Both commercial and academic LS tools (Brayton & Mishchenko, 2010) employ sophisticated human-designed heuristics to obtain approximate solutions, often resulting in sub-optimal outcomes. Traditional approaches address this problem by following the generate-then-optimize paradigm from two distinct perspectives: boolean functions (Lai et al., 1993; Nabulsi et al., 2017) and and-inverter graphs (Brayton, 2006; Bertacco & Damiani, 1997; Mishchenko et al., 2011). Recent research (Belcak & Wattenhofer, 2022; Petersen et al., 2022) suggests that neural methods, which directly generate circuit graphs, carry the promising potential to simultaneously generate and optimize circuits, thereby producing compact initial solutions for subsequent optimization. On the other hand, symbolic regression (SR) has shown great success in recovering underlying mathematical functions from datasets, demonstrating its effectiveness in various scientific discovery tasks.

Figure 1: The duality between boolean function discovery and mathematical function discovery.

We conclude that synthesizing boolean functions from input-output examples (i.e., a truth table) is analogous to recovering mathematical functions from a dataset in the boolean domain, as illustrated in Figure 1. First, the truth table corresponds to the dataset. Second, the logical operations AND, OR, and NOT correspond to the arithmetic operations of multiplication, addition, and subtraction, respectively. Finally, boolean expression trees correspond to mathematical expression trees.

Thus, a desired question is: Can we leverage the strong capabilities of SR methods to recover not only exact but also compact boolean functions for significantly boosting logic synthesis?

In this paper, we first investigate whether existing SR methods can effectively learn boolean functions. Through extensive experiments, we found that these SR methods struggle to recover *exact* and *compact* boolean functions for logic synthesis from a given truth table. The primary challenges include: (1) the underlying boolean functions are significantly more complex than typical mathematical functions, (2) a significant gap exists between the complexity of boolean functions and the size of synthesized circuits, and (3) the multi-objective nature to achieve not only exact recovery but also compact generation towards circuit optimization.

To address these challenges systematically, we propose a novel approach called the **s**ymbolic fac**tor**ized boolea**n** s**e**archer (SINE) to recover exact and compact boolean functions for logic synthesis. The key innovations of SINE are as follows. (1) **Factorized Boolean Function Representation**. Inspired by the Shannon decomposition theorem (Gdanskiy et al., 2020), SINE factorizes the original boolean function into multiple simplified sub-functions while preserving exact functionality. This significantly reduces the complexity of the underlying functions, thereby improving the recovery accuracy. (2) **Self-Symmetric Tree Search**. Based on the key observation that logical sharing is significant for reducing circuit sizes, SINE proposes to encourage search self-symmetric tree expressions with the goal of maximizing subexpression sharing, which can be implemented using shared nodes in the synthesized circuit. (3) **Lexicographic Optimization**. Given the multi-objective nature of learning boolean functions, SINE incorporates the lexicographic optimization technique into our tree search, which prioritizes accuracy over circuit size.

We evaluate SINE on three widely-used circuit benchmarks. Experiments demonstrate that SINE generates boolean functions more accurately than the general mathematical SR methods, achieving significant improvement in terms of accuracy. Moreover, we compare SINE with two EDA-based LS methods and experiments show that SINE significantly outperforms the baselines in terms of circuit size. Our results demonstrate the strong capability of our SINE to learn exact and compact boolean functions toward circuit synthesis.

We summarize our contributions as follows. (1) We empirically show that existing SR methods struggle to effectively learn boolean functions, and provide key insights for the challenge. (2) To the best of our knowledge, SINE is *the first* symbolic search framework that can exactly recover optimized boolean functions towards circuit optimization, opening a new direction towards neural circuit synthesis with emerging symbolic regression techniques. (3) SINE is a novel Boolean Searcher framework that addresses the challenges of boolean function learning systematically. (4) Experiments demonstrate that SINE significantly outperforms competitive baselines, significantly improving the accuracy and decreasing circuit size by up to 24.32%.

## 2 RELATED WORK

**Machine Learning for Logic Synthesis (LS)** Traditional LS methods synthesize a circuit from a given truth table via manually designed heuristics, such as sum-of-products (Nabulsi et al., 2017)

and binary decision diagrams (Lai et al., 1993). Recently, many researchers investigate machine learning for LS (Rai et al., 2021; Belcak & Wattenhofer, 2022; Schmitt et al., 2021; 2023), which offers promising approaches to learn to generate compact circuits with smaller sizes. Specifically, they formulate the input-output pairs in a truth table as a training dataset, and leverage machine learning methods to generate a circuit fitting the dataset. Roughly speaking, these approaches fall into two categories as follows. (1) In the early stages of the International Workshop on Logic & Synthesis (IWLS) competition, researchers proposed to use decision trees and random forests to generate circuits from the input-output pairs of specified truth tables. (2) In recent IWLS competitions, researchers have proposed using deep differentiable logic gate networks (Petersen et al., 2022) to generate circuits from the complete input-output pairs of specified truth tables.

**Symbolic Regression (SR)** SR aims to recover underlying analytical expressions from given datasets, which has shown great success in many scientific discovery tasks. Roughly speaking, existing SR methods fall into three categories as follows. (1) Genetic programming (GP) based SR approaches (Espejo et al., 2009; Virgolin et al., 2021; He et al., 2022) maintain a population of expression "individuals" that evolve using genetic operators such as selection, crossover, and mutation. While GP-based approaches can be effective, it tends to struggle to scale to large-scale SR problems. (2) In recent years, transformer-based SR methods (Biggio et al., 2021; Kamienny et al., 2022; Holt et al., 2023; d'Ascoli et al., 2023) have been shown successfully recovering large-scale mathematical expressions with up to twelve input variables. However, they suffer from high training costs and poor generalization performance. (3) In contrast, the reinforcement learning (RL) and monte-carlo tree search (MCTS) based SR methods (Petersen et al., 2019; Sun et al., 2022; Xu et al.) formulates the expression generation problem as a RL problem, and uses RL and/or MCTS methods to solve the problem. They have achieved state-of-the-art performance on multiple SR benchmarks.

## 3 BACKGROUND

**Problem Formulation of Learning in Logic Synthesis (LS)** In recent years, synthesizing circuits from truth tables via machine learning has gained increasing attention (Rai et al., 2021; Belcak & Wattenhofer, 2022; Schmitt et al., 2021; 2023). Given a truth table $\mathcal{T}$, we assume it describes a boolean function $f : \mathbb{B}^n \to \mathbb{B}^m$ for a circuit with $n$ input and $m$ output. In terms of the truth table, each line in the truth table represents an input-output pair $(\mathbf{x}, \mathbf{y})$, indicating the output signals $\mathbf{y}$ produced by the circuit for the given input signals $\mathbf{x}$, where $\mathbf{x} \in \mathbb{R}^n$ and $\mathbf{y} \in \mathbb{R}^m$. Then the input-output pairs in the truth table constructs a dataset $\mathcal{D} = (\mathbf{X}, \mathbf{Y}) = \{(\mathbf{x}_i, \mathbf{y}_i)\}_{i=1}^{2^n}$. Given the dataset, we aim to learn a boolean function $\hat{f} : \mathbb{B}^n \to \mathbb{B}^m$ that precisely fits the dataset. Based on the learned boolean function, we can easily construct a corresponding circuit.

**Symbolic Regression (SR) for Learning Boolean Functions** SR aims to find a mathematical expression $f$ to best fit a given dataset $\mathcal{D}$. To this end, many existing SR approaches represent any mathematical expression by an algebraic expression tree, where internal nodes are operators (e.g., $+, \times, \sin$) and terminal nodes are input variables and/or constants. We assume $\tau = [\tau_1, \ldots, \tau_n]$ is a pre-order traversal of such an expression tree. Note that there is a one-to-one correspondence between an expression tree and its pre-order traversal. Each $\tau_i$ is an operator, input variable, or constant selected from a library of possible tokens, e.g., $[+, -, \times, \div, \sin, \cos, \exp, \log.x]$. To apply the existing SR methods to learning boolean functions, we reset the operators as $and, or, not$, and the library of possible tokens at each step as $[and, or, not, x_1, \ldots, x_n]$.

## 4 KEY CHALLENGES IN SYMBOLIC REGRESSION FOR LOGIC SYNTHESIS

### 4.1 SCALABILITY CHALLENGE: EXPONENTIAL GROWTH OF UNDERLYING FUNCTIONS

To evaluate whether existing symbolic regression (SR) methods can recover exact boolean functions for logic synthesis (LS), we evaluate four popular SR methods on circuits from three widely-used benchmarks (Lowd & Domingos, 2012; Boucher & King, 2010; He et al., 2021). The four SR methods include GPLearn (Espejo et al., 2009), Boolformer (d'Ascoli et al., 2023), DSR (Petersen et al., 2019), and SPL (Sun et al., 2022). For fairness, we implemented these methods by replacing the mathematical operators with basic boolean operators, while keeping other implementations unchanged. Please refer to Appendix A for more implementation details.

**Poor Scaling** Previous work (Holt et al., 2023) has shown the existing SR methods are able to recover mathematical expressions with up to twelve input variables. However, as shown in Figure

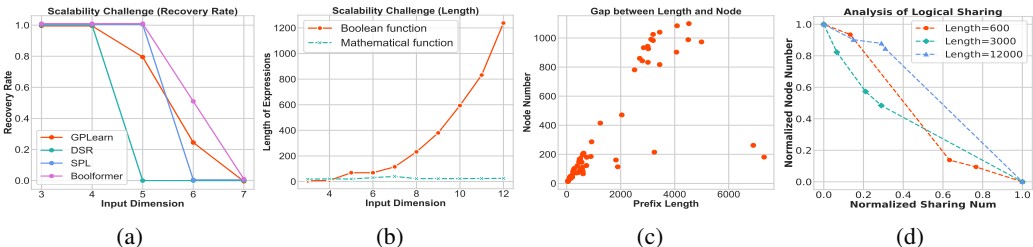

(a)  (b)  (c)  (d)

Figure 2: (a) The existing SR methods struggle to recover exact boolean functions when the input dimension exceeds seven. (b) The length of boolean functions exponentially grows with the input dimension. (c) The length of boolean functions is not positively correlated with the synthesized circuit size. (d) The number of logical sharing significantly impacts the circuit size.

2a, the existing SR methods struggle to recover exact boolean functions when the input dimension exceeds seven, while the input variables of real circuits are often larger than seven. This poses scalability challenge, which severely hinders the application of SR methods to circuit synthesis.

**Exponential Growth of Underlying Boolean Functions** To further analyze the poor scaling phenomenon, we compare the length of mathematical functions with boolean functions from real circuit benchmarks. As shown in Figure 2b, the results demonstrate that the length of boolean functions exponentially grows with the input dimension. This significantly expands the search space, making it challenging for existing SR methods to accurately recover boolean functions We discuss two major reasons for this exponential growth problem as follows. First, the functionality of real circuits, such as arithmetic and control circuits, are often complex. Second, unlike the general mathematical function space which includes advanced symbolic operators such as $\sin$, $\cos$, and $\exp$, the boolean function space primarily consists of simple boolean operators $and, or, not$. Consequently, it requires a large number of fundamental boolean operators and variables to express complex circuits.

## 4.2 COMPLEXITY CHALLENGE: LOGICAL SHARING SIGNIFICANTLY MATTERS

For the task of LS, it not only requires recovering exact boolean functions but also finding compact functions for circuit optimization. However, we empirically show the traditional complexity measure of boolean functions is inconsistent with the size of synthesized circuits. The major reason for this inconsistency stems from the neglectness of logical sharing in boolean functions.

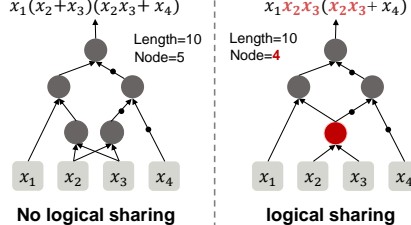

Figure 3: Illustration of Logical Sharing

**Gap between Function Complexity and Circuit Size** As shown in Figure 2c, we analyze real circuits from widely-used circuit benchmarks and found a significant gap between the lengths of their corresponding boolean functions and circuit sizes. The results demonstrate that the complexity of boolean functions is not the sole factor affecting circuit size.

**Logical Sharing Significantly Matters** Logical sharing refers to the logic nodes or sub-expressions that are shared by multiple logical components. As illustrated in Figure 3, $ab$ is a logical sharing node that appears twice in the expressions and can be shared as a node in the final circuit. To demonstrate that the number of logical sharing nodes significantly impacts the circuit size, we conduct the following experiments on real circuits. The results in Figure 2d indicate that, when controlling for Boolean functions of the same length, the circuit size tends to be inversely proportional to the number of logical sharing. This suggests that incorporating logical shares into Boolean expressions can effectively reduce the circuit size. Overall, learning an accurate boolean function with both low complexity and increased logical sharing is beneficial for generating high-quality circuits. Due to limited space, we defer more detailed results in Appendix C.1.

## 5 SYMBOLIC FACTORIZED BOOLEAN SEARCHER

In this section, we present our proposed **s**ymbolic factor**i**zed boolea**n** **se**archer (SINE) framework. As shown in Figure 4, we first present an overview of our proposed SINE.

To address the scalability challenge mentioned in Section 4.1, we propose a factorized Boolean function representation inspired by the Shannon decomposition theorem (Gdanskiy et al., 2020). As shown in Figure 5, we first decompose the original complex Boolean function (truth table) into mul-

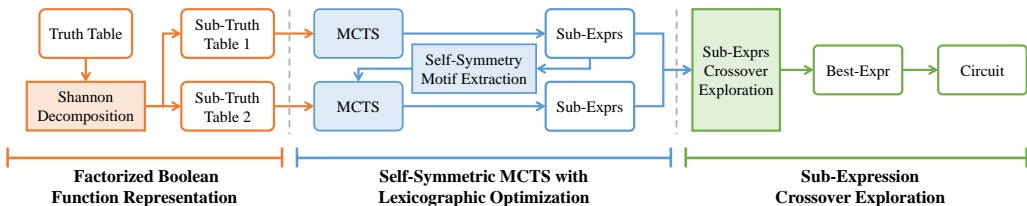

Figure 4: A simple illustration of our SINE framework.

tiple simple sub-functions (sub-truth tables) to reduce the search space by selecting several variables to decompose the function. For simplicity, we design a greedy strategy for variable selection.

To address the complexity challenge mentioned in Section 4.2, we propose a Self-Symmetric Tree Search (STC) framework with lexicographic optimization inside it. As shown in Figure 5, STC consists of multiple symmetrical tree search agents, each of which finds a set of compact boolean functions for each decomposed sub-truth table. Based on the key observation that logical sharing significantly matters, STC introduces symmetry into the boolean function learning across different tree search agents to maximize the logical sharing across different found boolean functions. Moreover, STC incorporates the lexicographic selection inside the tree search to tackle the multi-objective problem. Finally, STC applies a genetic crossover to the generated boolean sub-expressions to further optimize the final synthesized circuit.

## 5.1 FACTORIZED BOOLEAN FUNCTION REPRESENTATION

As shown in Figure 2b, we found that the exponential growth in the length of boolean functions significantly expands the search space and increases search difficulty. To address this challenge, we propose a Factorized Boolean Function Representation method motivated by the Shannon decomposition theorem (Gdanskiy et al., 2020) to decompose the original boolean function into several simplified sub-functions. Specifically, SINE iteratively selects a variable $X_i$ to decompose a given boolean function $f$ of $n$ input variables into two sub-functions with $n-1$ input variables, i.e.,

$$f(X_1, X_2, \ldots, X_n) = X_i \cdot f_1(X_1, \ldots X_i = 1, \ldots, X_n) + X_i' \cdot f_2(X_1, \ldots, X_i = 0, \ldots, X_n).$$

Then, we can further decompose the sub-functions $f_1$ and $f_2$ by selecting another variable.

In practice, the underlying compact boolean function is unknown, while we have the truth table containing the complete input-output pairs of a given circuit. Nevertheless, each boolean function can be represented by a unique truth table. Thus, we apply the aforementioned decomposition mechanism of boolean functions to truth tables. Specifically, given a truth table $\mathcal{T}$ with $n$ input and $m$ output, we decompose it into two sub-tables $\mathcal{T}_1$ and $\mathcal{T}_2$, where $\mathcal{T}_1$ and $\mathcal{T}_2$ contain a half of input-output pairs in $\mathcal{T}$ with the $i$-th input being 1 and 0, respectively.

To implement the aforementioned decomposition mechanism, an appropriate variable selection policy is required. As shown in Figure 7b in Appendix C, experiments demonstrate that different selected variables can lead to significantly variable final circuit sizes. For simplicity, we design a greedy selection approach, which greedily select the variable based on the circuit size after decomposition. We leave learning a variable selection policy as future work.

## 5.2 SELF-SYMMETRIC TREE SEARCH WITH LEXICOGRAPHIC OPTIMIZATION

**Dataset Formulation** Given a truth table $\mathcal{T}$ with $n$ input and $m$ output, we first decompose it into $2^k$ factored sub-tables by recursively selecting $k$ variables to obtain $2^k$ sub-tables with $n-k$ input and $m$ output. Each sub-table is formulated as $m$ training datasets. For each dataset, we aim to learn a compact boolean function that can precisely fit the dataset. Thus, the learning problem is formulated as learning boolean functions from the decomposed $m \times 2^k$ datasets with the goal of minimizing the final synthesized circuit size.

**Boolean Symbolic Regression Formulation** Any boolean expression can be represented by a combinatorial set of symbols and boolean operators, and further expressed by a parse tree structure (Hopcroft et al., 2006; Kusner et al., 2017). Following (Sun et al., 2022), we use a tuple $G = (V, \Sigma, R, P)$ to represent the expression tree, where $V$ denotes a finite set of non-terminal nodes corresponding to the independent variables (e.g., $x_0, x_1$), $\Sigma$ a finite set of terminal nodes, $R$ a reward for a given node, $P$ a finite set of production rules. Each production rule is interpreted as a

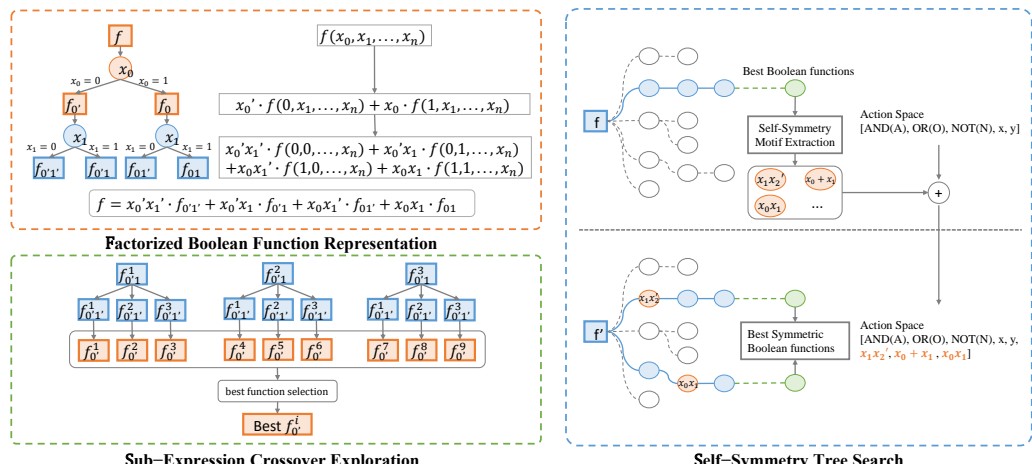

Figure 5: Our SINE consists of three main components to recover exact and compact boolean functions for logic synthesis. Please see Section 5 for details.

mapping from a single non-terminal symbol in $V$ to one or multiple terminal/non-terminal node(s) in $(V \cup \Sigma)^*$ where $*$ represents the Kleene star operation (Piao & Salomaa, 2012). In the tree search procedure, we define the action space $A = P$ and the state space $S$ as all possible traversals of production rules selected in ordered sequences. In our boolean function learning task, the production rules include basic boolean operators AND, OR, and NOT. As shown in Figure 4, the goal of our STC is to find a boolean expression tree that maximizes the expected reward of root node $f$.

**Self-Symmetric Tree Search** Given the factored sub-truth tables, the problem can be formulated as a boolean symbolic regression problem. Due to the scalability challenge of the boolean function as shown in Figure 2a, we apply the Monte-Carlo Tree Search (MCTS) algorithm, which is indeed well-suited for handling SR problems with vast search space (Sun et al., 2022), to search for the boolean functions. However, different from the traditional single-objective MCTS, our problem involves complex objectives with not only exact recovery but also expression optimization towards circuit optimization. To address this problem, we propose a Self-Symmetric Tree Search (STC) framework with Lexicographic Optimization.

*Progressively Expanded Libraries via Motif Mining* As shown in Figure 2d, we found that the Logical sharing significantly matters with the circuit size. Moreover, we observe that the shared logical node in the circuit indeed corresponds to symmetric sub-expressions in the circuit's corresponding boolean function as shown in Figure 3. Thus, to integrate this prior information into our search process, we propose a Self-Symmetric Tree Search framework with the goal of learning as many as possible symmetric sub-expressions during the learning process of the boolean function. To this end, we leverage the idea of motif learning for adaptively mining a series of motifs (i.e., sub-expressions) from those already searched boolean functions. Then, by adding these mined motifs into the action space, the subsequent tree search process can implicitly search boolean expressions with many sub-expressions symmetric with previous searched expressions, thus leading to many logical sharing nodes in the final circuit.

More specifically, we present details on our adaptive motif learning mechanism as follows. As shown in Figure 4, given a pair of symmetric sub-truth tables, the small function structures are adaptively extracted as motifs from one of the generated boolean functions and included in the action space of its symmetric tree search agent. For example, the motifs extracted from a boolean function $(x_0 + x_1) \times ((!x_0 + x_2) + x_1)$ are $(x_0 + x_1)$ and $(!x_0 + x_2)$. Selecting these small function structures is based on the observation that they typically function as deeper nodes in the generated circuit graph, leading to increased logical sharing. Consequently, these extracted motifs effectively guide the search process and ultimately generate symmetric sub-expressions with more shared structures.

*Lexicographic Selection* In our problem, due to the multi-objective optimization, the rewards received for a given state is a vector $\mathbf{R} = (R_a, R_c)$, where $R_a$ is defined as the accuracy of the simulation boolean expression tree and $R_c$ is the expression's corresponding circuit size. However,

different from the typical multi-task tree search method where multiple objectives can be optimized simultaneously without any priority relationships, our problem places a strong emphasis on the accuracy metric over the complexity metric. Therefore, we incorporate Lexicographic Optimization into our STC framework, which prioritizes the optimization of the accuracy metric before considering the complexity metric. Specifically, we integrate Lexicographic Optimization into the selection step. Following (Kocsis & Szepesvári, 2006), in each selection step, the STC agent maintains a trade-off between exploration and exploitation by selecting actions that maximize the Vectorized Upper Confidence Bounds applied for Trees (UCT), formulated as:

$$\mathbf{UCT}(s, a) = \mathbf{Q(s, a)} + c \frac{\ln[N(s)]}{N(s, a)} \tag{1}$$

Here, $\mathbf{Q}(s, a)$ is a two-dimensional vector that represents the average rewards of playing action $a$ in state $s$ in the simulations performed in the history, encouraging the exploitation of the current best child node; $N(s)$ is the number of times state $s$ has been visited, and $N(s, a)$ is the number of times action $a$ has been selected at state $s$. Given the two-dimensional vector $UCT$, we choose the action based on lexicographic optimization. Specifically, lexicographic optimization first identifies the action set that maximizes the first term of the $UCT$. Then, within this selected action set, the selected action is that maximizes the second term. Due to limited space, we defer the details of our lexicographic selection algorithm in Appendix B.3.1. Overall, our proposed lexicographic selection enables STC to effectively recover exact and compact boolean sub-functions.

### 5.3 POST-GENERATION CROSSOVER

If the generated boolean expression fails to exactly recover the given sub-truth table, we design a legalization mechanism to improve its accuracy to 100%. Then, our STC generates multiple compact boolean sub-functions for each sub-truth table, all characterized by the same accuracy but potentially varying in complexity. To further optimize the final synthesized circuit, we propose a post-generation crossover mechanism by recombining sub-functions from different sub-truth tables. This process is divided into two main steps. In the first step, we apply crossover to the decomposed sub-expressions for each output. In the second step, we conduct an additional crossover among the expressions of each output for multi-output circuits. To manage the potentially overwhelming number of permutations resulting from numerous output bits, we utilize sampling during this crossover. Eventually, we select the function with the highest accuracy and the smallest circuit size as the final solution. For more details, please refer to Appendix B.3.2.

## 6 EXPERIMENTS

Our experiments consist of four main parts. (1) We evaluate the offline accuracy of our SINE on three widely-used open-source circuit benchmarks. (2) We evaluate the online circuit size of our SINE. (3) We perform carefully designed ablation studies to provide further insight into SINE. (4) We perform visualization experiments and explainability analysis.

**Benchmarks** We evaluate our approach on three widely-used logic synthesis circuit benchmarks—Arithmetic (Lowd & Domingos, 2012), Espresso (Boucher & King, 2010), LogicNets (He et al., 2021). We selected five, three, and two circuits from each dataset. The input of these circuits ranges from five to twelve with outputs varying from one to thirteen.

**Experimental Setup** Throughout all experiments, we use ABC (Brayton & Mishchenko, 2010) as the backend LS framework, which is a state-of-the-art open-source LS framework and is widely used in research of machine learning for LS. Once generating a boolean function, we apply the default command *write_eqn* to transform the boolean function to a logic circuit. We employ MCTS as our search framework for the factored truth tables. Different from the normal symbolic function space based on operators addition, subtraction, and multiplication, we apply boolean operators $and, or, not$, as the fundamental symbolics for boolean function space. The decomposition times for every circuit is up to three.

**Competitive Baselines** Our baselines include four widely used pre-trained, evolutionary algorithmic, state-of-the-art (SOTA) learning-based SR approaches and three EDA-based methods. (1) **Boolformer** (d'Ascoli et al., 2023) is the first pre-trained based SR method applied for boolean function learning. (2) **GPLearn** (Espejo et al., 2009) is a classical evolutionary approach for symbolic regression. (3). **SPL** (Sun et al., 2022) is a SOTA MCTS-based symbolic regression method.

Table 1: The offline results demonstrate that our SINE significantly improves the accuracy and reduces the wrong bits of the generated circuits.

| Benchmark | | | GPLearn | | Boolformer | | DSR | | SPL | | SINE | |
|---|---|---|---|---|---|---|---|---|---|---|---|---|
| Circuit | PI | PO | Acc(%)↑ | Init Node↓ | Acc(%)↑ | Init Node↓ | Acc(%)↑ | Init Node↓ | Acc(%)↑ | Init Node↓ | Acc(%)↑ | Init Node↓ |
| Ci1 | 5 | 1 | 99.38 | 29 | 100 | 12 | 88.13 | 13 | 100 | 14 | 100 | 13 |
| Ci2 | 6 | 1 | 72.34 | 17 | 85.94 | 39 | 66.25 | 1 | 78.13 | 7 | 98.75 | 23 |
| Ci3 | 6 | 2 | 98.44 | 25 | 100 | 15 | 90.78 | 3 | 92.97 | 11 | 100 | 13 |
| Ci4 | 6 | 7 | 96.92 | 46 | 99.55 | 48 | 93.77 | 117 | 99.55 | 41 | 100 | 41 |
| Ci5 | 8 | 2 | 92.89 | 12 | 99.22 | 37 | 92.93 | 6 | 88.28 | 14 | 100 | 25 |
| Ci6 | 9 | 4 | 90.82 | 11 | 94.14 | 125 | 90.18 | 4 | 93.55 | 19 | 98.44 | 61 |
| Ci7 | 9 | 13 | 84.82 | 67 | 96.86 | 175 | 75.84 | 61 | 92.37 | 80 | 97 | 127 |
| Ci8 | 10 | 10 | 87.42 | 37 | 99.38 | 101 | 87.95 | 17 | 94.69 | 66 | 100 | 60 |
| Ci9 | 12 | 3 | 92.94 | 8 | 95.44 | 29 | 90.69 | 6 | 95.48 | 24 | 97.14 | 123 |
| Ci10 | 12 | 3 | 87.93 | 6 | 95.43 | 18 | 88.39 | 6 | 94.65 | 21 | 98.44 | 148 |
| average | | | **90.39** | 25.8 | **96.60** | 59.9 | **86.49** | 23.4 | **92.97** | 29.7 | **98.98** | 63.4 |

Table 2: We legalize every generated circuit, i.e., make the circuit's accuracy 100%, for an intuitive and fair comparison. The results demonstrate that SINE significantly outperforms all baselines in terms of the legalized initial circuit size.

| Benchmark | | | GPLearn | | Boolformer | | DSR | | SPL | | SINE | |
|---|---|---|---|---|---|---|---|---|---|---|---|---|
| Circuit | PI | PO | Acc(%)↑ | Init Node↓ | Acc(%)↑ | Init Node↓ | Acc(%)↑ | Init Node↓ | Acc(%)↑ | Init Node↓ | Acc(%)↑ | Init Node↓ |
| Ci1 | 5 | 1 | 100 | 29 | 100 | 12 | 100 | 29 | 100 | 14 | 100 | 13 |
| Ci2 | 6 | 1 | 100 | 70 | 100 | 66 | 100 | 66 | 100 | 65 | 100 | 41 |
| Ci3 | 6 | 2 | 100 | 25 | 100 | 15 | 100 | 31 | 100 | 14 | 100 | 13 |
| Ci4 | 6 | 7 | 100 | 61 | 100 | 53 | 100 | 153 | 100 | 45 | 100 | 41 |
| Ci5 | 8 | 2 | 100 | 46 | 100 | 44 | 100 | 59 | 100 | 47 | 100 | 25 |
| Ci6 | 9 | 4 | 100 | 162 | 100 | 138 | 100 | 174 | 100 | 192 | 100 | 72 |
| Ci7 | 9 | 13 | 100 | 231 | 100 | 172 | 100 | 281 | 100 | 248 | 100 | 116 |
| Ci8 | 10 | 10 | 100 | 103 | 100 | 105 | 100 | 72 | 100 | 122 | 100 | 60 |
| Ci9 | 12 | 3 | 100 | 572 | 100 | 556 | 100 | 698 | 100 | 565 | 100 | 173 |
| Ci10 | 12 | 3 | 100 | 597 | 100 | 587 | 100 | 592 | 100 | 608 | 100 | 172 |
| average | | | 100 | **189.6** | 100 | **174.8** | 100 | **215.5** | 100 | **192** | 100 | **72.6** |

(4) **DSR** (Petersen et al., 2019) is a SOTA RL-based method. In terms of EDA-based methods, we apply two heuristics named **SOP** and **BDD**. Please refer to appendix A for more details.

**Evaluation Metrics** Throughout all experiments, we evaluate our method in two separate phases, i.e., the offline and online phases. (1) In the offline phase, we evaluate the accuracy (higher is better) of the generated logic circuit and the number of its wrong bits. The accuracy is defined as the ratio of correctly predicted output bits number to the total number of output bits. The wrong bits refer to the disparity between the total number of output bits and the number of correctly predicted output bits in the generated logic circuits, which show the accuracy difference more intuitively. (2) In the online phase, we evaluate our approach from two perspectives: the initial size of the generated circuit and the final size of the post-optimization circuit. Initial size denotes the number of nodes of the generated circuit, which significant impacts the chip area. Moreover, to demonstrate that the generated logic circuit serves as a well-founded initial solution for subsequent optimization, we evaluate the number of nodes of the circuit optimized by synthesis operators.

**Experiment 1. The Offline Evaluation: Comparison with SR Methods** To demonstrate the superiority of SINE, we compare SINE with four SR baselines on ten real circuits across a wide range of input sizes. The results in Table 1 demonstrate that our SINE significantly outperforms all baselines in terms of accuracy. Specifically, the number of circuits with 100% accuracy generated by our SINE is 2.5 times that of the state-of-the-art Boolformer. To ensure the exactness of our generated initial circuits, we design a simple legalization method. After legalization, the results in Table 2 show that SINE significantly reduces the initial circuit size compared to the four SR methods. Overall, the offline results demonstrate that SINE can accurately discover compact boolean functions, thus significantly boosting logic synthesis. We defer more results to Appendix C.2.

**Experiment 2. The Online Evaluation: Comparison with EDA-based Methods** In this subsection, we evaluate both the online initial size, i.e., the number of circuit nodes, and the online final size of the generated circuits. The results in Table 3 indicate that our SINE method significantly outperforms all baselines in terms of circuit size. Specifically, SINE achieves an improvement of up to 20% in initial size. Moreover, considering the post-optimization results, our method shows an average increase of 10.10% in optimized nodes compared to the default operators, indicating that our method provides a robust initial solution for post-circuit optimization. Overall, the online results demonstrate that SINE can precisely recover optimized Boolean functions for circuit optimization, thus achieving a significant reduction in circuit area. Please refer to Appendix C.3 for more results.

**Experiment 3. Ablation Study** To understand the contribution of the main components in SINE, we perform an ablation study on four diverse circuits from widely-used benchmarks. Our method

Table 3: The online results demonstrate the strong ability of our method to recover compact boolean functions for subsequent circuit optimization. We apply the Resyn2 operator on the initial circuit.

| Benchmark | | | SOP | | BDD | | SINE | | | |
|---|---|---|---|---|---|---|---|---|---|---|
| Circuit | PI | PO | Init Node↓ | Opt Node↓ | Init Node↓ | Opt Node↓ | Init Node↓ | Impr(%) | Opt Node↓ | Impr(%) |
| Ci1 | 5 | 1 | 15 | 12 | 15 | 12 | 13 | 13.33 | 10 | 16.67 |
| Ci2 | 6 | 1 | 46 | 40 | 43 | 39 | 41 | 4.65 | 37 | 5.13 |
| Ci3 | 6 | 2 | 15 | 12 | 15 | 12 | 13 | 13.33 | 12 | 0.00 |
| Ci4 | 6 | 7 | 43 | 24 | 44 | 24 | 41 | 6.82 | 24 | 0.00 |
| Ci5 | 8 | 2 | 28 | 23 | 29 | 23 | 25 | 13.79 | 20 | 13.04 |
| Ci6 | 9 | 4 | 82 | 68 | 78 | 66 | 72 | 7.69 | 67 | -1.52 |
| Ci7 | 9 | 13 | 147 | 116 | 145 | 111 | 116 | **20.00** | 84 | **24.32** |
| Ci8 | 10 | 10 | 82 | 61 | 74 | 61 | 60 | 18.92 | 54 | 11.48 |
| Ci9 | 12 | 3 | 236 | 206 | 206 | 177 | 173 | 16.02 | 149 | 15.82 |
| Ci10 | 12 | 3 | 194 | 151 | 190 | 138 | 172 | 9.47 | 116 | 15.94 |
| average | | | 88.80 | 71.30 | 83.90 | 66.30 | 72.60 | **12.40** | 57.30 | **10.09** |

Table 4: The ablation study on several diverse circuits. The results demonstrate that each component in SINE plays an important role in improving accuracy and reducing circuit size.

| Method | Ci1 | | Ci2 | | Ci6 | | Ci7 | |
|---|---|---|---|---|---|---|---|---|
| | Acc(%)↑ | Nodes↓ | Acc(%)↑ | Nodes↓ | Acc(%)↑ | Nodes↓ | Acc(%)↑ | Nodes↓ |
| MCTS | 100 | 14 | 78.13 | 7 | 93.55 | 19 | 92.37 | 80 |
| F | 100 | 18 | 93.75 | 46 | 97.75 | 80 | 94.22 | 159 |
| FM | 100 | 18 | 98.75 | 41 | 98.44 | 72 | 97 | 146 |
| FMC | 100 | 13 | 98.75 | 23 | 98.44 | 61 | 97 | 127 |
| SINE (Ours) | 100 | 13 | 100 | 41 | 100 | 72 | 100 | 116 |

comprises four main components: Factorized Boolean Function Representation(**F**), Self-Symmetry motif learning (**M**), Genetic Crossover (**C**), and Legalization (**L**). The results in Table 4 demonstrate that each component in SINE(=FMCL) plays an important role in improving the accuracy and size of the generated circuits. First, **F** outperforms MCTS on accuracy, demonstrating that factoring the truth table significantly reduces the learning difficulty. Second, **FM** outperforms **F** on circuit size, demonstrating the superiority of the self-symmetry motif learning. Third, **FMC** further improves **FM**, showing that the genetic crossover is important for the optimization of the generated circuit. Finally, our SINE ensures legalized circuit generation while maintaining little size increase through the legalization method. Due to limited space, please refer to Appendix C.4 for more details.

**Experiment 4. Visualization and Explainability Analysis** To provide further insight into the boolean function learned by SINE, we visualize the circuit Ci3 and its corresponding boolean functions generated by SINE and the traditional SOP. Moreover, we provide statistic results for explainability analysis in Appendix C.5. These results suggest the following. (1) The boolean functions generated by SINE possess significantly more logical sharings than SOP, which significantly reduces the circuit size. (2) As shown in Figure 6, small boolean structures, such as $(!x_2 * x_3)$, $(!x_1 * x_5)$, correspond to deeper nodes in the generated circuit, which are more likely to be shared. This demonstrates the effectiveness of our proposed motif mining strategy.

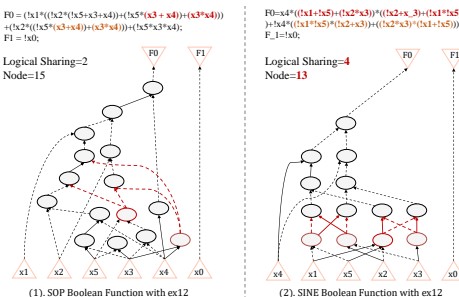

Figure 6: The visualization results demonstrate SINE's strong ability to capture more logical sharing, leading to smaller circuits.

# 7  CONCLUSION

In this paper, we propose SINE, a novel approach for recovering exact and compact Boolean functions for logic synthesis. SINE includes a factorized Boolean function representation to reduce the search space and a self-symmetric tree search framework. Compared to standard methods, SINE achieves a remarkable average reduction of 49.02% in the wrong bits and decreases circuit size by up to 24.32%. Our experiments show that there is significant potential for enhancing our current search strategy. In the future, we plan to incorporate more powerful search methods, such as large language models (LLMs), and extend our framework to more logic synthesis tasks.

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

Yilong Xu, Yang Liu, and Hao Sun. Reinforcement symbolic regression machine.

## A  IMPLEMENTATION DETAILS OF THE BASELINES

Below, we provide short descriptions of the four SR baseline methods and two heuristic boolean optimization methods.

- **GPLearn**: GPLearn Espejo et al. (2009) provides an efficient and rapid GP-based SR implementation. However, despite its speed, it may exhibit instability and poor scalability.

- **Boolformer**: Boolformer d'Ascoli et al. (2023) is a pre-trained method that first applies transformer framework into Boolean symbolic discovery.

- **DSR**: DSR Petersen et al. (2019) is a search-based method that employs a gradient-based risk-seeking RL approach combined with a recurrent neural network (RNN) to generate a probability distribution over expressions.

- **SPL**: SPL Sun et al. (2022) is a search-based symbolic regression method that employs a Monte Carlo tree search (MCTS) agent to explore optimal expression trees based on measurement data. SPL is one of the SOTA SR method.

- **SOP**: SOP Nabulsi et al. (2017) is a heuristic method integrated into the widely-used logic synthesis framework ABC. This method achieves precise logic synthesis by representing truth tables as sums of products.

- **BDD**: BDD Lai et al. (1993) is another heuristic method included in the ABC logic synthesis framework. It achieves precise logic synthesis by applying the Shannon decomposition theorem and representing the truth table as a binary decision diagram.

## B  IMPLEMENTATION DETAILS ON THE FACTORIZED BOOLEAN FUNCTION REPRESENTATION

### B.1  HARDWARE SPECIFICATION

Our experiments were executed on a Linux-based system equipped with a 3.60 GHz Intel Xeon Gold662 6246R CPU and NVIDIA RTX 3090 GPU.

### B.2  IMPLEMENTATION DETAILS ON THE FACTORIZED BOOLEAN FUNCTION REPRESENTATION

#### B.2.1  THE HEURISTIC DECOMPOSITION POLICY

To evaluate whether the decomposition variable impacts the circuit size, we designed a decomposition variable selection rule called **RandomHeuristics**, which randomly selects a decomposition variable for each sub-truth table. We evaluated RandomHeuristics on two randomly generated circuits, D1 and D2, as well as two real circuits, D3 and D4. We follow the d'Ascoli et al. (2023) to generate two small circuits with up to twenty nodes. The real circuits are chosen from the circuit benchmarks with up to ten inputs. Using the open-source state-of-the-art logic synthesis (LS) framework ABCNabulsi et al. (2017) as the backend, we assessed the synthesis performance by the number of nodes in the generated circuit. RandomHeuristics was evaluated on each circuit over ten random seeds. Each bar in Figure 7b shows the mean and standard deviation (stdev) of its performance on each circuit. As shown in Figure 7b, the performance of RandomHeuristics on each circuit varies widely depending on the decomposition variable.

### B.3  IMPLEMENTATION DETAILS ON SELF-SYMMETRIC TREE SEARCH

#### B.3.1  LEXICOGRAPHIC SELECTION

Due to the complex objectives with not only exact recovery but also expression optimization towards circuit optimization in our problem, we incorporate Lexicographic Optimization inside the selection of our Tree Search framework. The detailed algorithm is presented in Algorithm 1.

---

**Algorithm 1** Compute a lexicographic maximum.

---

**Require:** Current state s, Set $A \subseteq \mathbb{R}^n$ (n=2 in our problem).
1: **for** k = 1, ..., n **do**
2:      Find an solution $a^{(k)}$ to the optimization problem:

$$\underset{a \in A}{\text{maximize}} \quad Q_k(s, a)$$

$$\text{subject to} \quad Q_i(s, a) \geq Q_i(s, a^{(k-1)}) \quad \text{for all } i \in [k-1]$$

3: **end for**
4: **return** $a^{(n)}$

---

### B.3.2 SUB-FUNCTION CROSSOVER

In real-world circuits, the number of outputs is not always one. Therefore, a full traversal for crossover would be impossible. To address this problem, we employ a random sampling policy for the combination process. Specifically, we randomly select one expression from each sub-function set and combine them, repeating this process 10,000 times. From these 10,000 combinations, we choose the function with the highest accuracy and minimal circuit size as the final solution.

## C MORE RESULTS

### C.1 MOTIVATING RESULTS

To further investigate the impact of Boolean expression complexity and logical sharing on circuit size, we selected five circuits for each logical sharing value. As shown in Figure 7a, we found that when controlling for the same number of logical sharing, the length of the Boolean expression is proportional to the circuit size. The results further confirm that logical sharing and complexity are two key factors influencing circuit size.

### C.2 MORE RESULTS OF OFFLINE EVALUATION

More results about the Offline Evaluation can be found in Tables 5. The results demonstrate that our method outperforms all SR methods on circuits with large outputs. Specifically, our SINE achieves an improvement of up to 1100 wrong bits.

### C.3 MORE RESULTS OF ONLINE EVALUATION

More results about the Online Evaluation can be found in Tables 7 and 8. In Table 7, we apply dc2 as the optimization operator and we found that SINE achieves an average improvement of 12.40% of Initial circuit size and 10.10% of Optimized circuit size. Moreover, results in table 8 demonstrate that our SINE outperforms all baselines in Initial circuit size and Optimized circuit size. Overall, we can conclude that our method is capable of recovering compact boolean functions for LS.

### C.4 MORE RESULTS OF ABLATION STUDIES

In this part, we present more ablation results in Table 9. Specifically, we conduct the ablation experiments on eight test circuits and the results demonstrate that each component in SINE plays an important role in improving accuracy and reducing circuit size.

### C.5 MORE RESULTS OF VISUALIZATION

In this part, we present the statistic results for explainability analysis in Table 6. The results demonstrate that our SINE are capable of capturing more logical sharing than heuristics baseline, and thus synthesising smaller circuit. Moreover, we present more visualization results of the boolean functions generated by our method and heuristics method SOP. As shown in Figure 8 and 9, we

present the length of boolean functions and its corresponding circuit size. The visualization results demonstrate that our SINE learns a more compact boolean function with more logical sharing.

## D LICENSE

The code and model will be publicly accessible. We use standard licenses from the community. We include the following licenses for the codes, datasets and models we used in this paper.

**datasets**:

- Arithmetic: Arithmetic
- Espresso: Espresso
- LogicNets: LogicNets

**Codes**:

- GPLearn:GPLearn
- DSR: BSD-3-Clause
- Boolformer: Boolformer
- SPL:SPL

**Models**:

- Boolformer:Boolformer

Table 5: More offline results on large output circuits

| Benchmark | | | GPLearn | | Boolformer | | DSR | | SPL | | SINE | | |
|---|---|---|---|---|---|---|---|---|---|---|---|---|---|
| Circuit | PI | PO | Acc(%)↑ | Wrongs.↓ | Acc(%)↑ | Wrongs.↓ | Acc(%)↑ | Wrongs.↓ | Acc(%)↑ | Wrongs.↓ | Acc(%)↑ | Wrongs.↓ | Impr(%) |
| Ci11 | 9 | 79 | 96.40 | 1455 | 98.81 | 480 | 96.50 | 1417 | 97.32 | 1085 | **99.09** | 368 | 23.33 |
| Ci12 | 12 | 3 | 93.83 | 759 | 95.36 | 571 | 93.86 | 754 | 94.61 | 663 | 96.13 | 476 | 36.87 |

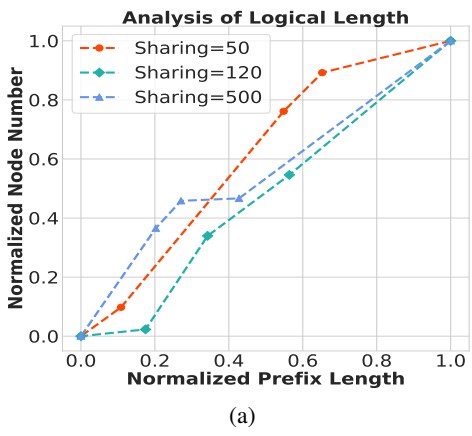

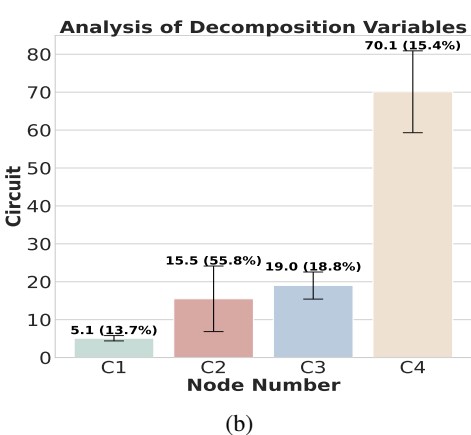

(a)                                         (b)

Figure 7: (a). The results demonstrate that the complexity of Boolean functions significantly impacts the circuit size. (b). The results demonstrate that the selection of the decomposition variable significantly impacts the circuit size.

Table 6: We provide statistics for the boolean functions generated by SOP, BDD, and our SINE, including the length of boolean functions, the number of Logical Sharing, and the initial circuit size.

| Metrics | SOP | BDD | SINE |
|---|---|---|---|
| Length | 457.6 | 449.4 | 453.6 |
| Logical Sharing | 14.2 | 18.70 | 48.80 |
| Init Node | 88.80 | 83.90 | 72.60 |

Table 7: The online results demonstrate the strong ability of our method to recover compact boolean functions for circuit optimization. We apply the dc2 operator on the initial circuit.

| Benchmark | | | SOP | | BDD | | SINE | | | |
|---|---|---|---|---|---|---|---|---|---|---|
| Circuit | PI | PO | Init Node↓ | Opt Node↓ | Init Node↓ | Opt Node↓ | Init Node↓ | Impr(%) | Opt Node↓ | Impr(%) |
| Ci1 | 5 | 1 | 15 | 12 | 15 | 12 | 13 | 13.33 | 10 | 16.67 |
| Ci2 | 6 | 1 | 46 | 40 | 43 | 38 | 41 | 4.65 | 35 | 7.89 |
| Ci3 | 6 | 2 | 15 | 12 | 15 | 12 | 13 | 13.33 | 12 | 0.00 |
| Ci4 | 6 | 7 | 43 | 22 | 44 | 20 | 41 | 6.82 | 21 | -5.00 |
| Ci5 | 8 | 2 | 28 | 25 | 29 | 23 | 25 | 13.79 | 20 | 13.04 |
| Ci6 | 9 | 4 | 82 | 63 | 78 | 63 | 72 | 7.69 | 65 | -3.17 |
| Ci7 | 9 | 13 | 147 | 105 | 145 | 109 | 116 | **20.00** | 83 | **23.85** |
| Ci8 | 10 | 10 | 82 | 59 | 74 | 55 | 60 | 18.92 | 48 | 12.73 |
| Ci9 | 12 | 3 | 236 | 204 | 206 | 172 | 173 | 16.02 | 148 | 13.95 |
| Ci10 | 12 | 3 | 194 | 127 | 190 | 138 | 172 | 9.47 | 109 | 21.01 |
| average | | | 88.80 | 66.90 | 83.90 | 64.20 | 72.60 | **12.40** | 55.10 | **10.10** |

Table 8: The online results demonstrate the strong ability of our method to recover compact boolean functions for circuit optimization. We apply the compress2 operator on the initial circuit.

| Benchmark | | | SOP | | BDD | | SINE | | | |
|---|---|---|---|---|---|---|---|---|---|---|
| Circuit | PI | PO | Init Node↓ | Opt Node↓ | Init Node↓ | Opt Node↓ | Init Node↓ | Impr(%) | Opt Node↓ | Impr(%) |
| Ci1 | 5 | 1 | 15 | 12 | 15 | 12 | 13 | 13.33 | 10 | 16.67 |
| Ci2 | 6 | 1 | 46 | 40 | 43 | 38 | 41 | 4.65 | 37 | 2.63 |
| Ci3 | 6 | 2 | 15 | 12 | 15 | 12 | 13 | 13.33 | 12 | 0.00 |
| Ci4 | 6 | 7 | 43 | 24 | 44 | 24 | 41 | 6.82 | 28 | -16.67 |
| Ci5 | 8 | 2 | 28 | 23 | 29 | 23 | 25 | 13.79 | 20 | 13.04 |
| Ci6 | 9 | 4 | 82 | 68 | 78 | 67 | 72 | 7.69 | 68 | -1.49 |
| Ci7 | 9 | 13 | 147 | 110 | 145 | 109 | 116 | **20.00** | 84 | **22.94** |
| Ci8 | 10 | 10 | 82 | 61 | 74 | 61 | 60 | 18.92 | 54 | 11.48 |
| Ci9 | 12 | 3 | 236 | 206 | 206 | 176 | 173 | 16.02 | 149 | 15.34 |
| Ci10 | 12 | 3 | 194 | 148 | 190 | 139 | 172 | 9.47 | 116 | 16.55 |
| average | | | 88.80 | 70.40 | 83.90 | 66.10 | 72.60 | **12.40** | 57.80 | **8.05** |

Table 9: The ablation study that involves more circuits. The results demonstrate that each component in SINE plays an important role in improving accuracy and reducing circuit size.

| Method | Ci1 | | Ci2 | | Ci3 | | Ci4 | |
|---|---|---|---|---|---|---|---|---|
| | Acc(%)↑ | Nodes↓ | Acc(%)↑ | Nodes↓ | Acc(%)↑ | Nodes↓ | Acc(%)↑ | Nodes↓ |
| MCTS | 100 | 14 | 78.13 | 7 | 93.97 | 15 | 99.56 | 41 |
| F | 100 | 18 | 93.75 | 46 | 100 | 17 | 100 | 48 |
| FM | 100 | 18 | 98.75 | 41 | 100 | 15 | 100 | 41 |
| FMC | 100 | 13 | 98.75 | 23 | 100 | 13 | 97 | 41 |
| SINE (Ours) | 100 | 13 | 100 | 41 | 100 | 13 | 100 | 41 |
| Method | Ci5 | | Ci6 | | Ci7 | | Ci8 | |
| | Acc(%)↑ | Nodes↓ | Acc(%)↑ | Nodes↓ | Acc(%)↑ | Nodes↓ | Acc(%)↑ | Nodes↓ |
| MCTS | 89.29 | 33 | 93.55 | 19 | 92.37 | 80 | 95.69 | 66 |
| F | 100 | 33 | 97.75 | 80 | 94.22 | 159 | 100 | 74 |
| FM | 100 | 25 | 98.44 | 72 | 97 | 146 | 100 | 69 |
| FMC | 100 | 25 | 98.44 | 61 | 97 | 127 | 100 | 60 |
| SINE (Ours) | 100 | 25 | 100 | 72 | 100 | 116 | 100 | 60 |

Table 10: We compare our SINE method with four symbolic regression baselines across five test circuits (i.e., Ci1-5). The results show that our approach generates smaller circuits without high-performance hardware and significant time costs.

| Method | runtime/training time(average, s) | hardware requirements | legalization init nd (average) |
|---|---|---|---|
| SINE (Ours) | 999.85 | CPU | **72.6** |
| SPL | 1845.82 | CPU | 192 |
| DSR | 1982.08 | CPU | 215.5 |
| boolformer | 3 days | **GPU** | 174.8 |
| GPLearn | **50.95** | CPU | 189.6 |

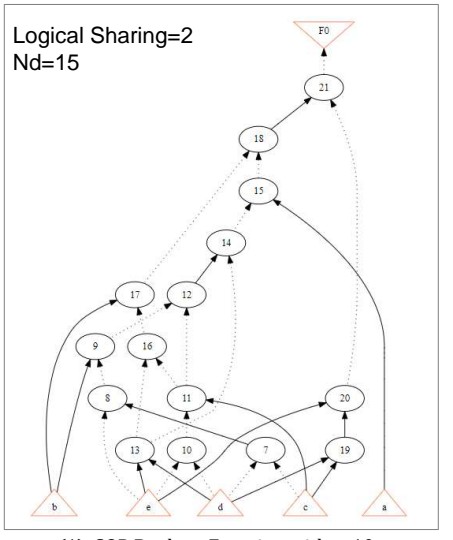
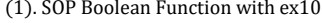

(1). SOP Boolean Function with ex10

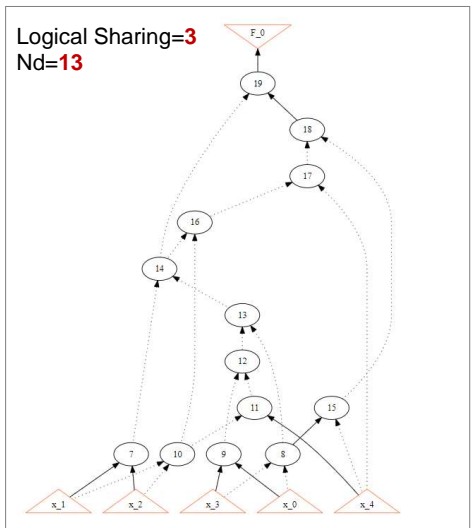

(2). SINE Boolean Function with ex10

Figure 8: The visualized circuit generated by SOP and our SINE on circuit Ci1.

Table 11: We compare our approach with four SR baselines on parital support noisy circuits. The inactive PI refers to the number of inactive variable we add to the original circuits. The results demonstrate that our method outperforms all of the baselines.

| Benchmark | | | | SINE (Ours) | GPlearn | Boolformer | DSR | SPL |
|---|---|---|---|---|---|---|---|---|
| Circuit | PI | PO | Inactive PI | Acc(%) | Acc(%) | Acc(%) | Acc(%) | Acc(%) |
| Ci11 | 12 | 3 | **1** | 96.13 | 93.83 | 95.36 | 93.86 | 94.61 |
| Ci12 | 12 | 6 | **2** | 99.20 | 97.51 | 99.20 | 95.62 | 97.61 |
| Ci13 | 16 | 13 | **7** | 97.96 | 95.19 | 96.30 | 90.93 | 96.30 |
| **Average** | | | | **97.76** | 95.51 | 96.95 | 93.47 | 96.17 |

Table 12: Comparison of our approach with a Decision Tree based method from IWLS 2020 on ten test circuits. The results demonstrate that our method achieves an average improvement of 57.64% than the DT approach on the legalized circuits.

| | SINE | DT | SINE_legalize | DT_legalize |
|---|---|---|---|---|
| Acc(average, %) | 98.98 | 99.89 | 100 | 100 |
| Init nd | 63.4 | 193.8 | **72.6** | 236.1 |

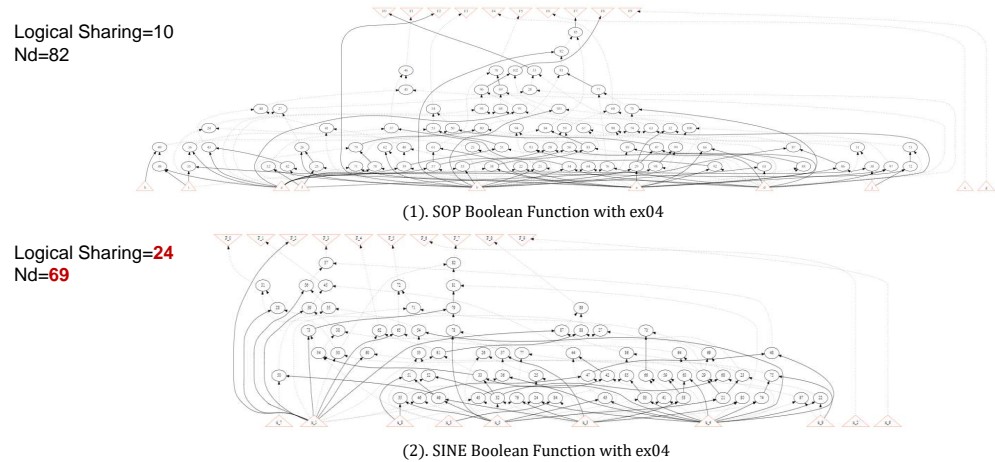

Logical Sharing=10
Nd=82

(1). SOP Boolean Function with ex04

Logical Sharing=**24**
Nd=**69**

(2). SINE Boolean Function with ex04

Figure 9: The visualized circuit generated by SOP and our SINE on circuit Ci8.

