# OpenReview forum: "Can Symbolic Regression of Boolean Functions Boost Logic Synthesis?"
_ICLR.cc/2025/Conference — ICLR 2025 Conference Withdrawn Submission_

### Official Review · Reviewer_DTvJ · 2024-11-01

**Soundness:** 1
**Presentation:** 2
**Contribution:** 1
**Rating:** 3
**Confidence:** 2

**Summary:**

The paper investigates symbolic regression (SR) as a tool for logic synthesis. Traditional SR methods, while effective for mathematical functions, have limitations in efficiently recovering complex Boolean. The authors introduce the "Symbolic Factorized Boolean Searcher" (SINE), an SR framework designed specifically for Boolean functions to address these limitations. Inspired by Shannon’s decomposition theorem, SINE decomposes complex Boolean functions into simpler sub-functions. Self-Symmetric Tree Search aims to maximize subexpression sharing. A lexicographic optimization prioritizes accuracy over minimizing circuit size.
The paper's experiments indicate that SINE outperforms existing SR and Electronic Design Automation (EDA) approaches in terms of both accuracy and compactness, achieving up to a 24.32% reduction in circuit size. The paper concludes that SINE offers a promising SR approach for more efficient circuit design.

**Strengths:**

The paper addresses the application of symbolic regression (SR) to logic synthesis, a significant topic in hardware design and circuit optimization. This intersection of machine learning, symbolic regression, and logic synthesis could inspire further research and experimentation.

The paper introduces a multi-faceted approach incorporating techniques such as Shannon decomposition, Monte Carlo Tree Search (MCTS), and lexicographic optimization. This methodological blend may inspire alternative approaches to Boolean function learning and encourage the ML community to explore similar strategies.


By discussing accuracy and circuit size as dual objectives, the paper addresses a realistic aspect of logic synthesis. This multi-objective viewpoint may contribute positively to discussions about balancing different goals in optimization problems.

**Weaknesses:**

The method section is imprecise. For example, the use of the term mathematical functions. I do not follow the distinction between mathematical functions and boolean functions, as boolean functions are mathematical functions. Lines 176ff argue about the exponential growth of circuits, but the arguments are unclear, especially compared to “mathematical functions.” While the number of distinct circuits/boolean functions is exponential in the input dimension, the number of distinct mathematical functions for any input dimension is infinite. I do not follow the arguments in this section.

Figure 3 does not match its explanation in the text (line 200)

Obvious statements are phrased as findings, for example, that logical operators _and/or_ correspond to _multiplication/addition_ (lines 065ff). Or the fact that more logical sharing reduces circuit size (lines 203ff).

The paper does not explain how the so-called “legalization” works. What is the goal of this technique? How can the accuracy be simply improved to 100%?

Evaluating just 10 circuits provides a narrow view of SINE’s effectiveness, especially since the authors do not mention how and why these circuits were selected from the datasets. Evaluations on such a small number of samples (even if they would be representatively chosen) cannot safely support any conclusions.

The results are generally inconclusive. Previous methods seem to perform equally well on the selected circuits, with only slight advantages in specific settings on some of the circuits that are not representatively chosen. In Table 1, SINE achieves the highest average accuracy, although only by 2%, which is given only 10 circuits, which is probably not statistically significant. Furthermore, the circuit size is larger than that of all the methods compared.

The objective of Experiment 2 is not clear. How does this correspond to Symbolic regression?

Lastly, key results mentioned in the conclusion are not supported by the experiments (an average reduction of 49.02%) or are cherry-picking the result of a single circuit (up to 24.32%).

In summary, the paper is questionably motivated (scaling of boolean functions vs. mathematical functions), and their experiments are insufficient.

**Questions:**

see above

---

### Official Review · Reviewer_LofD · 2024-11-08

**Soundness:** 3
**Presentation:** 2
**Contribution:** 2
**Rating:** 5
**Confidence:** 4

**Summary:**

This paper proposes a method for Boolean circuit synthesis and optimization, targeted largely towards small circuits with up to 12 inputs at most. The ideas proposed include, at their core, factorized representations of larger circuits, identifying repeated motifsm and MCTS based  search for optimized implementations.

**Strengths:**

The overall empirical results of the paper on small benchmark circuits demonstrate its gains over existing methods. However, in fairness most comparisons are with methods that address somewhat different problems, except perhaps the comparisons with EDA tools.

**Weaknesses:**

Two main contributions of the paper, factorizing Boolean circuits and finding common subexpressions, are strikingly similar to standard reduced ordered binary decision diagram (ROBDD) methods that have been use for years in representing and optimizing Boolean circuits. ROBDDs are built using factorized expressions and "reduction" steps that automatically identify common subexpressions exploiting the canonical nature of ROBDDs. MCTS methods are adapted on top of Sun et al.'s MCTS based symbolic regression methods, incorporating multiple objectives and a final "crossover" step.  I have to confess that I found the paper/methods very hard to follow; in part the writing could be improved substantially---crossover for example, is described in a short paragraph and details in the appendix are sparse also. There's a body of work on genetic algorithms for symbolic regressions (https://ieeexplore.ieee.org/abstract/document/501943 for a start), and it's unclear to me how this work differs from these. Overall, a large number of different ideas from literature have been combined in this paper from a range of prior work resulting in empirical gains, but what's missing for me is a clear novel contribution.

**Questions:**

Could you please describe your proposed methods for factroing and common subexpression elimination differ from what ROBDDs do?

---

### Official Review · Reviewer_V9Sr · 2024-11-08

**Soundness:** 3
**Presentation:** 3
**Contribution:** 2
**Rating:** 5
**Confidence:** 5

**Summary:**

In this paper, the authors present a Symbolic method for improving the logic synthesis by generating more compact logic functions. In order to avoid the problem of previous symbolic searchers, the authors propose a factorized Boolean function representation (based on BDD using shannon decomposition) that allows then to be efficiently searched using the symbolic searcher. In addition the authors effectively employ logic sharing. Finally, the authors also show the usage of the motif mining approach, where the SINE searcher finds symmetric expressions for more effective sharing and leading to smaller circuits size.

**Strengths:**

- The novel combination of symbolic search with factorization and motif search
- Evaluation that includes comparison with other SR algorithms.

**Weaknesses:**

- SINE seems to have problems with larger circuits. This in particular concerns circuits with 20, 30 or so variables. It would be helpful how the method works or does not for such larger circuits.
- While the authors effectively compared the method with other SR algorithms, the benchmarks used have been used by in general a large variety of algorithms. It is necessary to have a more general evaluation approach.
- The benchmarks, while in classical Logic Synthesis the functions in benchmarks are known, here authors select sort of anonymous logic functions. By anonymous I mean that we do not know the exact function being synthesized such as an N-bit adder or simply a larger combinatorial function.  The complexity of a synthesizer is not always only based on the size of the functions but also on their complexity. For instance S-box functions in cryptography for being highly non-linear to maximize diffusion. Linear functions such as functions from a same NPN class have a very low cost between each other. It is necessary to know what function is being synthesized so that the method can be assessed properly.

**Questions:**

I would like to see explicit benchmarks. For instance the set of all functions in a benchmark set such as the EPFL logic synthesis benchmark and see their individual analysis and evaluation. Having a full set of benchmarks is like evaluating a classifier on a large dataset. Few selected functions only give a pin-point understanding of the software and prevent a larger and broader evaluation of its performance.

---

### Official Review · Reviewer_6EFz · 2024-11-10

**Soundness:** 1
**Presentation:** 2
**Contribution:** 1
**Rating:** 3
**Confidence:** 4

**Summary:**

The reviewer believe that the authors have misrepresented the problem it aims to solve. It seems to confuse two distinct problems in electronic design automation: learning a boolean function from a partial truth table (which involves generalizing to unseen entries) and logic synthesis (which is about optimizing a boolean circuit from a fully specified truth table). Since the truth table is fully specified in this case, symbolic regression is unnecessary; the focus should be on circuit optimization, a well-known problem in logic synthesis.

The authors' use of Shannon's decomposition and factorized boolean functions is not new—it’s been explored for decades, with techniques like Binary Decision Diagrams (BDD) already addressing similar challenges. The method they propose, using Monte-Carlo tree search for sub-expression optimization, seems more like a clever engineering solution than a novel contribution.

Additionally, the paper doesn't consider standard benchmarks (e.g., MCNC, Opencore), which would help contextualize the results. I recommend the authors clarify whether they’re addressing logic synthesis or learning boolean functions, as their current approach and terminology seem confused. I have put more details in the weakness section.

**Strengths:**

The reviewer feels that the paper's fundamental assumption and the problem statement itself is convoluted. All the reviewer can say is the authors have picked an open problem of logic synthesis in the EDA community however, the paper do not achieve its intended goal.

**Weaknesses:**

This paper has a fundamental flaw in posing the problem statement and the way authors have motivated the problem and approached the solution. Basically, in the electronic design automation community, there are two different problem statements: 1) Given a partial truth table, can one learn a boolean function which can generalize on the rest of the entries of the truth table with maximum accuracy. Rai et al. 2022 and Abbe et al. 2023 elaborately define this problem statement and attempt to solve this. A natural extension of this problem statement is: given the learned function, can one generate a boolean circuit and optimize the circuit. The major challenge of this problem is to achieve 100% accuracy on unseen truth table entries which itself is a difficult task and problem statement. The reason being: any two boolean function can have only one truth table entry different and rest all same, making this problem statement challenging.

On the other hand, logic synthesis is well known problem statement since early 70s and 80s. The problem statement is: given a completely specified truth table (i.e. all entries of truth table is known before hand), generate an optimized boolean circuit which implements the functionality. This requires no “learning” at all, rather this is an optimization problem where the challenge is how to represent the functionality using minimum number of basic boolean operators i.e. AND, OR and NOT gates. The major challenge in this problem comes from multiple input multiple output hardware designs which requires considerable amount of logic sharing (as author mentioned) to obtain minimized circuit implementation. Till date, AIG (and-inverter-gate) and MIG (majority inverter gate) are the sota boolean representations which helps in getting optimized circuit representation.


Now coming to the questions posed by the authors in the paper: “Can we leverage strong capabilities of SR methods to recover not only exact but compact boolean functions for significantly boosting logic synthesis?”  First of all, if authors are considering the problem of logic synthesis, then the entire truth table entries are available (this is also mentioned in Section 5.1 Line 247). So if this is the case, for what purpose is Symbolic regression method is used for, as there are no partial truth table where one has to “learn” the output values. All the truth table entries and corresponding outputs are available.

Secondly, the authors claim that they propose to solve the problem using “factorized boolean function representation” motivated by Shannon’s decomposition is an age-old technique explored in the logic synthesis community. The method was proposed by Bryant et. al in 1986 using the method called Binary Decision Diagrams (BDD) (Paper: Graph-Based Algorithms for Boolean Function Manipulation). The observation by the authors that the ordering of the variable against which factorization is done greatly impacts circuit size is also not a new problem. It is already well known problem called finding reduced-ordered binary decision diagram (RO-BDD) for a given boolean function.

Thirdly, what authors is trying to achieve in this paper is first decomposing the truth table on k variables and the sub truth table obtained on “n-k” variable inputs are used to “learn” a boolean function. Again, these sub-tables on “n-k” boolean variables are already have an output value, therefore can be early represented as a boolean circuit using SOP (sum-of-product) form or POS (product-of-sum) form. The review clearly don’t understand what the authors doing when they mention they are “learning” the boolean function considering this as a dataset. There is nothing to learn here as all inputs and outputs are known. What makes sense is: Given 2^k truth tables, how to represent them using common boolean subexpression so that circuit size can be minimized. Therefore, this becomes a sub-expression search problem across these sub truth tables, for which authors have used Monte-carlo tree search. However, this is limited novelty and more appropriate to be called an elegant engineering solution to an optimization problem.


Lastly, the authors have not at all considered well known benchmarks adopted in the literature for showing results for their solutions e.g. MCNC (1991), Opencore, EPFL arithmetic and control, OpenABC-D etc. These benchmarks have circuits ranging from 200 boolean gates to 400k boolean gates.

The reviewer advises the author to better present the problem statement on what they’re intending to solve: is it the logic synthesis or is it learning a boolean function from partial truth table? It appears that the authors are trying to solve the logic synthesis problem, however the reviewer believes using symbolic regression to “learn” circuit from sub truth-tables do not make sense and it is a simple common boolean expression search problem to achieve compact circuit.


References:

1) Abbe, Emmanuel, et al. "Generalization on the unseen, logic reasoning and degree curriculum." International Conference on Machine Learning. PMLR, 2023.

2) Rai, Shubham, et al. "Logic synthesis meets machine learning: Trading exactness for generalization." 2021 Design, Automation & Test in Europe Conference & Exhibition (DATE). IEEE, 2021.

3) Bryant, "Graph-Based Algorithms for Boolean Function Manipulation," in IEEE Transactions on Computers, vol. C-35, no. 8, pp. 677-691, Aug. 1986,

4) S. Yang, "Logic Synthesis and Optimization Benchmarks,” Technical Report, MCNC, Dec. 1988, published at 1989 MCNC International Workshop on Logic Synthesis

5) Amarú, L., Gaillardon, P. E., & De Micheli, G. (2015). The EPFL combinational benchmark suite. In Proceedings of the 24th International Workshop on Logic & Synthesis (IWLS)

6) Chowdhury, A. B., Tan, B., Karri, R., & Garg, S. (2021). Openabc-d: A large-scale dataset for machine learning guided integrated circuit synthesis. arXiv preprint arXiv:2110.11292

**Questions:**

This has been explained in detail in the weakness section.

---

### Note · Authors · 2024-11-25

I have read and agree with the venue's withdrawal policy on behalf of myself and my co-authors.